# The Importance of Healthy Habits to Compensate for Differences between Adolescent Males and Females in Anthropometric, Psychological and Physical Fitness Variables

**DOI:** 10.3390/children9121926

**Published:** 2022-12-08

**Authors:** Adrián Mateo-Orcajada, Lucía Abenza-Cano, Ana Cano-Martínez, Raquel Vaquero-Cristóbal

**Affiliations:** 1Facultad de Deporte, UCAM Universidad Católica de Murcia, 30107 Murcia, Spain; 2IES José Luis Castillo Puche, 30510 Yecla, Spain; 3Kinanthropometry International, UCAM Universidad Católica de Murcia, 30107 Murcia, Spain

**Keywords:** adolescents, basic psychological needs, body composition, life satisfaction, Mediterranean diet, physical activity, physical fitness, gender differences

## Abstract

Adolescence is a crucial stage in human development, and differences in psychological, physical and body composition variables between males and females have been amply demonstrated. However, the role played by certain healthy habits, such as the practice of physical activity, adherence to the Mediterranean diet (AMD) or the maintenance of an adequate weight status, in compensating for the differences found between males and females in these variables, is not well known. For this reason, the study aimed to analyze whether the practice of physical activity, optimal AMD, and adequate weight status can compensate for the differences between adolescent males and females in anthropometric variables, psychological state, and physical fitness. The sample was composed of 791 adolescents (404 males and 387 females) aged twelve to sixteen years old, whose anthropometric, psychological (autonomy, competence, relatedness, and life satisfaction), and physical fitness variables (cardiorespiratory fitness, upper strength and explosive lower limb power, hamstring and lower back flexibility, and speed) were measured. All measurements were carried out in a single day using the sports pavilion of the four participating schools. The most novel results of this research show that the practice of physical activity was determinant mainly in females, as it reduced the differences found in comparison with males in psychological (*p* < 0.001–0.045) and anthropometric variables (*p* < 0.001). Regarding weight status and AMD, these were still relevant for the adolescent population, mainly the achievement of optimal AMD, but males continued to present higher values in physical fitness tests (*p* < 0.001) and lower values in fat accumulation (*p* < 0.001), regardless of weight status or AMD. Thus, physical activity seems to be the most determining factor that compensates for the differences between adolescent boys and girls.

## 1. Introduction

Adolescence is a fundamental stage in the development of individuals and is characterized by physical, hormonal, and cognitive changes before reaching adulthood [1,2]. However, the timing of the onset of physical, hormonal, and cognitive changes differs between males and females, with females developing the earliest [3,4]. Moreover, the changes between males and females at this stage do not only differ in the time at which they occur, as hormonal differences during puberty are notable depending on the sex [2,5], giving rise to a phenomenon known as sexual dimorphism [6]. These differences in the maturation process between males and females during adolescence lead to changes in their behaviors related to physical activity [7], nutritional habits [8], and weight status [9], as well as in personal variables such as the anthropometry and derived variables [10], psychological state [11], or physical fitness [12].

With respect to the differences in the behavior of adolescent females and males, the regular practice of a physical activity is essential for the proper development of adolescents, as it influences their health during all stages of their lives, helping to prevent certain chronic diseases [13] and favoring the development of their physical abilities [14]. In addition, physical activity during adolescence plays a key role [15] in the maintenance of an active lifestyle during adulthood, as well as in the acquisition of other healthy lifestyle habits and the prevention of harmful lifestyle habits [16]. Furthermore, previous research has shown differences between males and females in the level of physical activity [7], with males practicing sports to a greater extent and with greater intensity than females [17].

Nutritional habits have also been considered one of the most determining factors in maintaining healthy habits [8,18]. These are some of the main elements to consider during adolescence, as previous research has shown that adolescents with poorer adherence to the Mediterranean diet (AMD) have a worse profile of plasmatic inflammation markers, along with the risk this poses for the development of metabolic syndrome, obesity, or insulin resistance [19], worse physical activity level [20], and worse mental health [21]. However, previous research has not shown differences between males and females, with similar levels of AMD observed in both genders [22,23].

In addition, weight status, and mainly, perceived weight status, are some of the most analyzed parameters in recent years due to their close relationship with health [24]. Obesity and overweight have been identified as the greatest pandemic of the 21st century, affecting an increasing number of children and adolescents at very early ages every year [25] and producing numerous non-communicable diseases such as diabetes, hypertension, hypercholesterolemia, asthma, cerebrovascular accidents, among others [9]. In this sense, it is relevant to know that the prevalence of obesity and overweight affects adolescent males and females differently, with males showing higher values than females regardless of the criteria used [26].

Previous scientific literature has shown how differences seem to be evident in the habits of adolescent males and females, to which we must also add the differences found, according to gender, in anthropometry and derived variables [10], psychological state [27], and physical fitness [10]. These changes occur because of the great hormonal changes between males and females from the peak of growth onwards, which result in sexual dimorphism [2,6] and mainly influence the sex differences in the accumulation of fat and muscle development [5,10]. Specifically, from this moment on, females tend to accumulate more fat [28], while males tend to have greater muscle development [10,29]. Considering that muscle development provides a competitive advantage for physical actions that depend on strength and power and that fat is a competitive disadvantage for actions that depend on endurance, agility, and power, biological differences could be the main cause of the differences in physical fitness that occur from this age onwards [28,29], mainly influencing endurance, strength, speed, agility, power, and flexibility, with all of these being greater for males, except in the case of flexibility, where the rapid increase in muscle mass experienced by males becomes a handicap for this ability, being greater in girls [10]. With respect to psychological state, the COVID-19 pandemic suffered in recent years had a great influence on adolescents [30], mainly on females, leaving them in a state of greater psychological vulnerability due to the decrease in well-being and behavioral changes, increasing their anxiety and depression to a greater extent [31]. In addition, the aforementioned body changes greatly influence self-concept and body image, with a particularly negative effect on females [32].

Thus, differences in anthropometry and derived variables between adolescent males and females have been noted in previous research [10]. In this sense, adolescent males present greater muscle development, while adolescent females accumulate a higher percentage of fat mass [10]. This difference is fundamental because of the relationship between the increase in fat mass and the decrease in muscle mass, an increased risk of certain diseases [33], and a decrease in physical performance [34].

Mental health problems are also a determining factor during adolescence due to their influence on the behavior of adolescents [35], which became more apparent after the COVID-19 pandemic, which led to significant changes in lifestyle habits and the way of relating [36]. It greatly affected the psychological state of adolescents [37,38], more noticeably and with greater risk for females [39]. However, the existing differences in certain psychological variables according to the gender of the adolescents are still unclear, as previous research in this area has not shown significant differences in life satisfaction between adolescent males and females [40]. In contrast, other studies have shown lower scores in this variable and competence in adolescent females [11,27].

In terms of physical fitness, the differences between adolescent males and females after the maturation period have been previously mentioned [10], with males performing better in strength and power tests while females scored higher in flexibility tests. These results acquire special importance because participation in certain physical activities is conditioned by the physical abilities relevant to these activities [41], giving rise to significant differences in sports participation between male and female adolescents. Males tend to participate to a greater extent, as most of the activities in which adolescents participate are collaborative-opposition group sports, such as soccer or basketball, where strength, speed, and power are decisive [42], while females see their sports practice reduced to individual and more aesthetic activities such as rhythmic gymnastics or swimming, decreasing their sports participation [42].

Therefore, previous scientific literature has shown that the healthy habits of adolescent boys and girls are different [7,8,9] and that there are significant differences between genders in anthropometric and derived variables [10], psychological state [27], and physical condition [10]. However, given this situation, no previous research has analyzed whether the lifestyle habits of adolescents are determinant in the changes produced in the anthropometric and derived variables, in psychological state and physical condition, and whether the differences between boys and girls could disappear in the anthropometric and derived variables, in psychological state and/or in physical condition when adopting a certain healthy habit. This leads to the following research question: is it possible that adolescent females, who normally have a greater accumulation of fat mass and less muscle mass development, who have more alterations in their psychological state, and who have a worse physical condition, can reduce, or even reverse the differences found with respect to males in these variables, by adopting a greater practice of physical activity, an optimal AMD and/or an adequate weight status?

This question leads to the following general aim for the present study: to analyze whether the practice of physical activity, optimal AMD, and adequate weight status can compensate for the differences between adolescent males and females in anthropometric and derived variables, psychological state, and physical fitness. In order to achieve this general aim, the following four specific objectives were proposed (a) to establish the differences between male and female adolescents in anthropometric and derived variables, psychological state, and physical fitness; (b) to determine whether the practice of physical activity can compensate for the differences found according to gender, in the anthropometric and derived variables, psychological state and physical fitness of the adolescents; (c) to analyze whether optimal AMD can compensate for the gender differences found in the anthropometric and derived variables, psychological state and physical fitness of the adolescents; and (d) to establish whether the maintenance of an adequate weight status can compensate for the differences found according to gender in the anthropometric and derived variables, psychological state and physical fitness of the adolescents.

Based on previous scientific literature and the objectives of the present investigation, the following hypotheses are proposed: (H1) males will show higher scores in all variables, except in those related to body fat and flexibility, compared to females; (H2) for females, the regular practice of physical activity will compensate for the differences in anthropometric and derived variables, psychological state and physical fitness with respect to males; (H3) for females, optimal AMD will compensate for the differences in anthropometric and derived variables, psychological state and physical fitness with respect to males; and (H4) for females, maintaining a normal weight will compensate for the differences in anthropometric and derived variables, psychological state and physical fitness with respect to males.

## 2. Materials and Methods

The research was carried out using a cross-sectional design, and the sample selection was performed with a non-probability convenience method. The STROBE statement was followed for the design of the research study and for drafting the manuscript [43]. In accordance with the World Medical Association and the guidelines of the Helsinki Declaration, the institutional ethics committee of the Catholic University of Murcia reviewed and approved the protocol before starting research (code: CE022102).

### 2.1. Research Model

Table 1 shows a summary of the dependent and independent variables included in the present research.

### 2.2. Participants

The sample consisted of adolescents enrolled in four schools in the Region of Murcia, selected from different geographical areas (north, south, east, and west), including the schools with the largest sample of adolescent students in compulsory secondary education, according to data from the Ministry of Education [44], thus achieving a representative sample of the Region of Murcia.

For the sample size calculation, the Rstudio 3.15.0 statistical software (Rstudio Inc., Boston, MA, USA) was used following the methodology from previous studies based on the standard deviation (SD) [45]. Thus, with an SD of 0.50 from previous research that examined the differences between adolescent males and females in the physical activity level [46] and with an estimated error (d) for a 99% confidence interval of 0.05, the minimum sample necessary for the conducting the research was 750 adolescents.

The final sample consisted of 791 adolescents (404 males and 387 females; mean age: 14.39 ± 1.26 years old; males’ mean age: 14.37 ± 1.25; females’ mean age: 14.40 ± 1.27). Participation in the study was voluntary, and all adolescents who met the following inclusion criteria were included: (a) attending compulsory secondary education; (b) completing all the measurements (questionnaires, anthropometric measurements, and physical fitness test); (c) participating in body composition measurements; (d) age between twelve and sixteen years old; and (e) not presenting any musculoskeletal injury or illness that would hinder participation in the physical tests or the completion of the questionnaires.

### 2.3. Instrumentation

#### 2.3.1. Questionnaire Measures

The questionnaires used in the present study were the “Physical Activity Questionnaire for Adolescents” (PAQ-A) [47,48], KIDMED [49], “Satisfaction with life scale” (SWLS) [50], and the “Basic Psychological Needs Scale” (BPNS) [51].

The PAQ-A, previously validated in Spanish [48], has an intraclass correlation coefficient of 0.71 and consists of nine items. The first eight are answered with a Likert scale of 1 to 5 points and allow information to be collected on the last seven days of physical activity. The last item allows for discovering if the subject could not perform physical activity in the week prior to the study. This questionnaire makes it possible to establish the physical activity status by discriminating between active and sedentary adolescents, obtaining a score between 1 and 5 according to the sum and subsequent averaging of the first eight items. For this purpose, a cut-off point of 2.75 was established, with active subjects being those above this value and sedentary those who were not [52].

To establish the AMD level, the “Mediterranean Diet Quality Index for children and adolescents” (KIDMED) questionnaire was used. This questionnaire was designed and validated with Spanish adolescents and showed moderate reliability and reproducibility values for use in adolescents (α = 0.79) [53]. It allows determining the level of AMD by means of 16 items rated by the adolescents with a score of 1 or 0, depending on whether the criterion was met. Subsequently, the scores obtained were totaled, considering that twelve of the questions had a positive connotation (+1) (favoring a good adherence), and four had a negative connotation (−1) (favoring an inadequate adherence). The final score was between 0 and 12 points, indicating the following adolescents’ AMD: poor adherence (0–3 points), need to improve adherence (4–7 points), or optimal adherence (8–12 points) [49].

The SWLS assesses the level of satisfaction with life by means of five items that are answered with a Likert scale of 1 to 5 points, with the final score ranging from 5 to 25 [50,54]. This scale was translated into Spanish and showed an adequate internal consistency (α = 0.84) for use with adolescents [54]. The BPNS assesses autonomy, competence, and relatedness by means of eighteen items (six items for each dimension) using a Likert scale from 1 to 6 points, with 6 being the minimum score and 36 being the maximum score in each dimension. The BPNS was previously translated and validated in Spanish [55]. This scale presents adequate external validity and internal consistency (competence = 0.80; autonomy = 0.69; and relatedness = 0.73) [51,55]. Therefore, both scales were used to assess life satisfaction and basic psychological needs, with both psychological constructs being of great relevance in the adolescent population, as previous research has shown their relationship with the acquisition of healthy habits such as physical activity [27], nutritional pattern [56], or weight status [57].

#### 2.3.2. Anthropometric Measurements

The anthropometric evaluation was composed of two basic measurements, body mass and height, performed using a TANITA BC-418-MA Segmental (TANITA, Tokyo, Japan) with an accuracy of 100 g and a SECA stadiometer 213 (SECA, Hamburg, Germany) with an accuracy of 0.1 cm, respectively; three skinfolds, triceps, thigh and calf, measured using a skinfold caliper (Harpenden, Burgess Hill, UK) with an accuracy of 0.2 mm; and five girths, arm relaxed, waist, hips, thigh, and calf, using an inextensible tape Lufkin W606PM (Lufkin, Missouri City, TX, USA) with a 0.1 cm accuracy for it measurements. All the instruments were previously calibrated, and all measurements were performed according to the protocol standardized by the International Society for the Advancement of Kinanthropometry (ISAK) [58]. The measurements were taken by anthropometrists (levels 2 to 4) accredited by the ISAK, following the protocol of this organization [58].

It should be noted that all the measurements corresponding to the same subject were performed by the same anthropometrist, who took a minimum of two measurements per variable, with a third measurement being necessary when the difference between the first two measurements was greater than 5% in the skinfolds and 1% in the rest of the measurements. The mean of the values when two measurements were taken, or the median when three measurements were taken, was used as the final value [58].

With the final values obtained from the anthropometric measurements, the BMI (kg/m^2^), Σ3 skinfolds (triceps, thigh, and calf), corrected arm girth [arm relaxed girth − (π × triceps skinfold)], corrected thigh girth [middle thigh girth − (π × thigh skinfold)] corrected calf girth [calf girth − (π × calf skinfold)], fat mass (%) [59], and muscle mass [60] were calculated.

The intra- and inter-evaluator technical error of measurements (TEM) were calculated in a sub-sample. The inter-evaluator TEM was 1.98% for skinfolds; 0.06% for the girths; and 0.03% for the basic measurements; and the intra-evaluator TEM was 1.21% for skinfolds; 0.04% for the girths; and 0.02% for the basic measurements.

#### 2.3.3. Physical Fitness Test

##### Cardiorespiratory Fitness

To assess the cardiorespiratory fitness of the adolescents, the 20-m shuttle run test was performed, in which the participants had to run 20 m as many times as possible following the incremental running beep set by a sound signal. The test ended when the adolescents were exhausted or when they did not run the 20 m in the time allowed. The last speed at which the adolescent finished the test was used to predict maximal oxygen consumption (VO_2_ max.) [61,62].

##### Strength and Power

To measure the upper body strength and the explosive lower limb power, the handgrip strength test [63] and the countermovement jump (CMJ) [64] test were used, respectively. The handgrip strength test consists of applying the maximum possible force on a Takei Tkk5401 digital handheld dynamometer (Takei Scientific Instruments, Tokyo, Japan) with the elbow fully extended, as this is the position in which maximum force is produced [65]. The CMJ consists of a vertical jump in which the aim is to reach the highest possible height. For this, the adolescent must maintain verticality throughout the flight phase and keep their hands on their hips. Initially, the subject stands with their hands on the waist, flexes their knees to 90, and performs a maximum knee extension [64]. To measure performance in this test, a force platform with a sampling frequency of 200 Hz (MuscleLab, Stathelle, Norway) was used.

##### Flexibility

The sit-and-reach test was used to assess hamstring and lower back flexibility. The participants were required to sit on the floor with their knees straight, legs together, ankles flexed at 90°, toes pointed upward, and the soles of the feet positioned flat against an Acuflex Tester III box (Novel Products, Rockton, IL, USA). From this initial position and with palms down, the subject performed a maximum trunk flexion, keeping the knees and arms fully extended, trying to reach the maximum possible distance by sliding the palms of the hands, one on top of the other, across the box [66].

##### Speed

For speed measurement, the 20-m sprint test was performed. To carry out this test, the adolescent had to run 20 m as fast as possible, after which the minimum time required to do so was recorded. At the start, the subject stood on the starting line and decided when to start the sprint [67]. To reduce the risk of the arms cutting the single-beamed photocells from 60% to 4%, the photocells were placed at hip height instead of chest height [68,69].

### 2.4. Procedure

First, the adolescents completed the questionnaires on physical activity level, AMD, life satisfaction, and satisfaction of basic psychological needs in a random order. Subsequently, anthropometric measurements were carried out. Once completed, an attempt at the sit-and-reach test was performed prior to the warm-up so that it would not influence the test result [70]. After completing this test, the adolescents were familiarized with the correct execution of the handgrip strength, CMJ, and 20-m sprint tests after an explanation was given. After familiarization, a warm-up was performed that included progressive running and joint mobility, after which the handgrip, CMJ, and 20-m sprint tests were performed twice. These tests were performed randomly for each adolescent, leaving two minutes of rest between each test attempt. Finally, the 20-m shuttle run test was performed. Five minutes of rest between runs were provided. A total of four researchers oversaw the familiarization and measurements of the tests. The researchers had previous experience in measuring physical fitness tests in adolescent populations. To avoid an inter-evaluator error in the assessments, the same researcher was responsible for measuring the same test during all the measurements.

All the measurements were performed on the same day, using the physical education class hour. The covered sports pavilions of the education centers were selected to reduce the polluting variables as much as possible. Given that the assessment of the physical fitness tests was carried out in a single measurement day, the metabolic demands of each test were considered, as well as the time needed for the subject to recover from the fatigue generated, by establishing the order of the tests according to the recommendations from the National Strength and Conditioning Association (NSCA) [71].

### 2.5. Data Analysis

The Kolmogorov-Smirnov normality test, as well as kurtosis, skewness, and variance, were used to analyze the normality of the data. Parametric tests were used because all of the variables had a normal distribution. First, descriptive statistics were performed for all the variables analyzed (mean values and standard deviation). Subsequently, a one-factor ANCOVA was conducted to determine the differences between adolescent males and females, with physical activity status, AMD, and weight status as covariates in the model. Three MANOVA analyses were performed. The first is to establish the differences between active and sedentary males and females; the second is to analyze the differences between normal weight and overweight/obese males and females; the third is to establish the differences between males and females with poor and optimal AMD. A pairwise comparison was then performed using the Bonferroni post-hoc test. The effect size calculation (η^2^) was performed according to previous research. Thus, the following values were utilized; small: ES ≥ 0.10; moderate: ES ≥ 0.30; large: ≥ 1.2; or very large: ES ≥ 2.0, with an error of *p* < 0.05 [72]. A value of *p* < 0.05 was set to determine statistical significance. The statistical analysis was performed with the SPSS statistical package (v.25.0; SPSS Inc., Chicago, IL, USA).

To clarify the statistical analysis, the adolescents were classified according to their level of physical activity, AMD, and weight status. Thus, the classification of males and females according to their level of physical activity was active males (AM), active females (AF), sedentary males (SM), and sedentary females (SF). Regarding AMD, the classification was poor adherence males (PDM), poor adherence females (PDF), optimal adherence males (ODM), and optimal adherence females (ODF). According to weight status, the classification was normal-weight males (NWM), normal-weight females (NWF), overweight/obese males (OWM), and overweight/obese females (OWF). It should be noted that in the AMD groups, adolescents with a poor AMD and need to improve AMD were pooled so that the sample size of this group would be similar when compared with optimal AMD, as with the weight status group in which overweight and obese adolescents were pooled so that the sample size would be as that of normal weight.

## 3. Results

### 3.1. Coefficient of Variation (CV) and Intraclass Correlation Coefficients (ICC) in Physical Fitness Test

The coefficient of variation (CV) and the intraclass correlation coefficients (ICC) were calculated the physical fitness tests that were repeated twice. The results are as follows: for the handgrip right arm test, ICC = 0.940, and CV = 3.14%; for the handgrip left arm test, ICC = 0.953, and CV = 3.10%; for the CMJ test, ICC = 0.892, and CV = 3.35%; and for the 20-m sprint test, ICC = 0.913, and CV = 2.15%.

### 3.2. Differences between Male and Female Adolescents in Anthropometric Variables, Psychological Variables, Physical Fitness Variables, Physical Activity Level, AMD, and Weight Status: Are There Differences between Male and Female Adolescents in These Variables (Specific Objective 1)?

Table 2 shows the differences between males and females in anthropometric variables, psychological variables, physical fitness variables, physical activity level, AMD, and weight status. Differences were found in all analyzed variables except for hips girth (*p* = 0.659), relatedness (*p* = 0.185), AMD (*p* = 0.540), and BMI (*p* = 0.083). Males showed higher scores in physical activity level (*p* < 0.001), life satisfaction (*p* = 0.008), competence (*p* < 0.001), and autonomy (*p* = 0.020), all physical condition variables (*p* < 0.001), except for the sit-and-reach test (*p* < 0.001); and all anthropometric variables (*p* < 0.001), except for the sum of 3 skinfolds (*p* < 0.001) and fat mass (*p* < 0.001). The inclusion of the physical activity status and the level of AMD in the model significantly affected the differences in relatedness (*p* < 0.001), with the rest of the variables maintaining the same significant differences. In addition, BMI had a significant effect on hip girth (*p* < 0.001), with the rest of the variables keeping the same significant differences, except for the case of autonomy, which ceased to be significant when considering this variable (*p* = 0.066).

### 3.3. Differences between Male and Female Adolescents with Different Physical Activity Status in Anthropometric Variables, Psychological Variables, Physical Fitness Variables, AMD, and Weight Status: Can the Practice of Physical Activity Compensate for the Differences Found According to Gender in the Anthropometric and Derived Variables, Psychological State and Physical Fitness of the Adolescents (Specific Objective 2)?

The differences between AM, AF, SM, and SF are shown in Figure 1. When comparing males with different levels of physical activity (AM vs. SM), significant differences were observed in competence (*p* < 0.001), autonomy (*p* = 0.011), and VO_2_ max. (*p* < 0.001), and AMD (*p* = 0.001), while for females with different levels of physical activity (PA vs. SF), the differences were significant in life satisfaction (*p* < 0.001), basic psychological needs (*p* < 0.001), and VO_2_ max. (*p* < 0.001), sit-and-reach (*p* = 0.007), CMJ (*p* < 0.001), 20-m sprint (*p* = 0.003), and AMD (*p* = 0.010), with active males and females showing higher scores in all the variables.

When comparing males and females with a similar level of physical activity (SM vs. SF and AM vs. AF), significant differences were observed in both active and inactive groups in all the variables analyzed, except for hip girth (SM vs. SF: *p* = 1.000; AM vs. AF: *p* = 1.000), autonomy (SM vs. SF: *p* = 1.000; AM vs. AF: *p* = 1.000), relatedness (SM vs. SF: *p* = 1.000; AM vs. AF: *p* = 1.000), AMD (SM vs. SF: *p* = 1.000; AM vs. AF: *p* = 1.000), and BMI (SM vs. SF: *p* = 1.000; AM vs. AF: *p* = 0.962), as well as life satisfaction (AM vs. AF: *p* = 1.000) and competence (AM vs. AF: *p* = 1.000) in the groups of active adolescents, in which no significant differences were found. The males in both groups showed higher scores in the anthropometric (SM vs. SF: *p* < 0.001; AM vs. AF: *p* < 0.001), psychological (SM vs. SF: competence *p* = 0.012, life satisfaction *p* = 0.021), and physical fitness (SM vs. SF: *p* < 0.001; AM vs. AF: *p* < 0.001) values, except for the sum of 3 skinfolds (SM vs. SF: *p* < 0.001; AM vs. AF: *p* < 0.001), fat mass (SM vs. SF: *p* < 0.001; AM vs. AF: *p* = 0.002) and sit-and-reach test (SM vs. SF: *p* < 0.001; AM vs. AF: *p* < 0.001), for which the females showed higher values.

Regarding the comparison of AM vs. SF, significant differences were found in all the analyzed variables, except for hip girth (*p* = 1.000) and BMI (*p* = 1.000), with AM showing higher scores (*p* < 0.001), except in the sum of 3 skinfolds (*p* < 0.001), fat mass (*p* < 0.001) and sit-and-reach test (*p* < 0.001); while for the AF vs. SM comparison, the differences were significant in all the variables, except for hip girth (*p* = 1.000), fat mass (*p* = 0.054), life satisfaction (*p* = 1.000), and BMI (*p* = 0.869), with the SM showing higher values in the anthropometric (*p* < 0.001) and physical fitness variables (*p* < 0.001), except for the sum of 3 skinfolds (*p* = 0.004) and-sit-and reach test (*p* < 0.001), while AF showed higher scores in basic psychological needs (competence: *p* < 0.001; autonomy: *p* = 0.045; relatedness: *p* = 0.036) and AMD (*p* = 0.001).

### 3.4. Differences between Male and Female Adolescents with Different AMD in Anthropometric Variables, Psychological Variables, Physical Fitness Variables, Physical Activity Level, and Weight Status: Can an Optimal AMD Compensate for the Gender Differences Found in the Anthropometric and Derived Variables, Psychological State and Physical Fitness of the Adolescents (Specific Objective 3)?

Figure 2 shows the differences between PDM, ODM, PDF, and ODF. The differences between males with different AMD (PDM vs. ODM) were significant in relatedness (*p* = 0.035) and physical activity level (*p* < 0.001), while in females with different AMD (PDF vs. ODF), the differences were significant in competence (*p* = 0.049) and physical activity level (*p* < 0.001), with males and females with optimal AMD showing higher scores.

Significant differences were also found when comparing males and females with the same AMD (PDM vs. PDF and ODM vs. ODF), finding significant differences in all the analyzed variables, except for hip girth (PDM vs. PDF: *p* = 0.604; ODM vs. ODF: *p* = 0.904), life satisfaction (PDM vs. PDF: *p* = 0.359; ODM vs. ODF: *p* = 0.331), autonomy (PDM vs. PDF: *p* = 0.294; ODM vs. ODF: *p* = 1.000), relatedness (PDM vs. PDF: *p* = 1.000; ODM vs. ODF: *p* = 0.616), and BMI (PDM vs. PDF: *p* = 1.000; ODM vs. ODF: *p* = 0.125), as well as competence in the optimal AMD group (ODM vs. ODF: *p* = 0.123), with males in both the poor and optimal AMD groups showing higher scores in all the variables analyzed, except for the sum of 3 skinfolds (*p* < 0.001–0.003), fat mass (*p* < 0.001–0.037) and sit-and-reach test (*p* < 0.001), in which females showed higher scores.

Regarding the comparison between PDF vs. ODM and PDM vs. ODF, the differences were significant between PDF vs. ODM in all the variables analyzed, except for hip girth (*p* = 1.000) and BMI (*p* = 0.430), while when comparing PDM vs. ODF, differences were found in all the variables analyzed, except for hip girth (*p* = 1.000), life satisfaction (*p* = 1.000), basic psychological needs (*p* = 1.000), physical activity level (*p* = 1.000), and BMI (*p* = 1.000), with the males, independently of their AMD, showing higher values in the anthropometric (*p* < 0.001), psychological (*p* < 0.001–0.012) and physical fitness (*p* < 0.001) variables, except for the sum of 3 skinfolds (*p* < 0.001), fat mass (*p* < 0.001) and sit-and-reach test (*p* < 0.001), where females, independently of their AMD, showed a higher score.

### 3.5. Differences between Male and Female Adolescents with Different Weight Statuses in Anthropometric Variables, Psychological Variables, Physical Fitness Variables, Physical Activity Level, and AMD: Can the Maintenance of an Adequate Weight Status Compensate for the Differences Found According to Gender in the Anthropometric and Derived Variables, Psychological State and Physical Fitness of the Adolescents (Specific Objective 4)?

Figure 3 shows the differences between NWM, NWF, OWM, and OWF. When comparing males with different weight statuses (NWM vs. OWM) and females with different weight statuses (NWF vs. OWF), the differences were significant in both groups in all analyzed variables (*p* < 0.001), except for height (NWM vs. OWM: *p* = 0.449; NWF vs. OWF: *p* = 1.000), life satisfaction (NWM vs. OWM: *p* = 0.839; NWF vs. OWF: *p* = 0.812), basic psychological needs (NWM vs. OWM: *p* = 1.000; NWF vs. OWF: *p* = 1.000), sit-and-reach test (NWM vs. OWM: *p* = 1.000; NWF vs. OWF: *p* = 1.000), physical activity level (NWM vs. OWM: *p* = 1.000; NWF vs. OWF: *p* = 1.000) and AMD (NWM vs. OWM: *p* = 1.000; NWF vs. OWF: *p* = 1.000), in addition to handgrip right arm (*p* = 0.107), handgrip left arm (*p* = 0.158) and 20-m sprint (*p* = 0.119) between NWF vs. OWF, with the OWM and OWF showing higher scores in all anthropometric variables (*p* < 0.001), as well as higher scores in the handgrip strength test in both arms in the OWM (*p* < 0.001).

Regarding the comparison between males and females with the same weight status (NWM vs. NWF and OWM vs. OWF), the differences were significant in all the variables analyzed between NWM and NWF, except for life satisfaction (*p* = 0.608), autonomy (*p* = 0.625), relatedness (*p* = 1.000), and AMD (*p* = 1.000), while in the comparison between OWM and OWF, the differences were also significant in all the variables analyzed, except for hip girth (*p* = 1.000), corrected calf (*p* = 0.205), fat mass (*p* = 0.115), basic psychological needs (*p* = 0.213–1.000), physical activity level (*p* = 0.272), and AMD (*p* = 1.000), with males in both groups obtaining higher scores in all the variables (*p* < 0.001), except for the sum of 3 skinfolds (*p* < 0.001), fat mass (*p* < 0.001) and sit-and-reach test (*p* < 0.001), for which the females showed higher values.

As for the differences between males and females with different weight statuses (NWM vs. OWF and NWF vs. OWM), it should be noted that differences were significant between NWM and OWF in all the variables analyzed, except for corrected girths (*p* = 0.198–1.000), life satisfaction (*p* = 0.131), basic psychological needs (*p* = 0.083–1.000), and AMD (*p* = 1.000), with the OWF showing higher scores in all anthropometric variables (*p* < 0.001), except for height (*p* < 0.001) and muscle mass (*p* = 0.002), while the NWM showed higher physical activity level (*p* = 0.004) and physical fitness test scores (*p* < 0.001), except for the sit-and-reach test (*p* < 0.001). The differences between NWF and OWM were significant in all variables except for life satisfaction (*p* = 0.085), basic psychological needs (*p* = 0.287–0.768), CMJ (*p* = 0.437), and AMD (*p* = 1.000), with the score of the OWM being higher in all the variables analyzed (*p* < 0.001), except for the sit-and-reach test (*p* < 0.001).

## 4. Discussion

The general aim of the present study was to analyze whether the practice of physical activity, optimal AMD, and adequate weight status could compensate for the differences between adolescent males and females in anthropometric and derived variables, psychological state, and physical fitness. In order to meet this general aim, the following four specific objectives were proposed (a) to establish the differences between male and female adolescents in anthropometric and derived variables, psychological state, and physical fitness; (b) to determine whether the practice of physical activity could compensate for the differences found according to gender in the anthropometric and derived variables, psychological state, and physical fitness of the adolescents; (c) to analyze whether optimal AMD could compensate for the gender differences found in the anthropometric and derived variables, psychological state, and physical fitness of the adolescents; and (d) to establish whether the maintenance of an adequate weight status could compensate for the differences found according to gender in the anthropometric and derived variables, psychological state, and physical fitness of the adolescents. The results of previous research lead us to pose the following hypotheses (H1) males will show higher scores in all the variables, except in those related to body fat and flexibility, compared to females; (H2) in females, the regular practice of physical activity will compensate for the differences in anthropometric and derived variables, psychological state, and physical fitness, with respect to males; (H3) in females, optimal AMD will compensate for the differences in anthropometric and derived variables, psychological state, and physical fitness, with respect to males; and (H4) in females, maintaining a normal weight will compensate for the differences in anthropometric and derived variables, psychological state, and physical fitness, with respect to males.

### 4.1. Differences between Male and Female Adolescents in Anthropometric Variables, Psychological Variables, Physical Fitness Variables, Physical Activity Level, AMD, and Weight Status: Are There Differences between Male and Female Adolescents in These Variables?

According to the first specific objective, to establish the differences between male and female adolescents in anthropometric and derived variables, psychological state, and physical fitness, the results of the present study are similar to those found in previous studies. Adolescent males showed a higher level of physical activity than females, in line with previous research [7,17]. With respect to physical fitness, males obtained higher scores in all the variables analyzed, except for the sit-and-reach test, corroborating the results of previous research [10,73]. As for the anthropometric variables, it was the females who obtained higher scores in the variables related to fat mass, highlighting the percentage of total fat or the sum of three skinfolds, while the males showed greater muscle mass and corrected girth, with these results being similar to those obtained previously in adolescents [10]. According to the psychological variables, males showed higher scores in life satisfaction, competence, and autonomy, consistent with previous research [11,27]. The differences found could be due to the existing sexual dimorphism between adolescent males and females, which leads to hormonal changes characterized by an increase in estrogen in girls, which favors fat accumulation and poorer sports performance, and in testosterone in males, with increases the muscle mass that facilitates performance in strength and power tests [2,6]. These results corroborate the first hypothesis (H1) put forward in the research, in that males would show higher scores in all variables, except those related to body fat and flexibility, compared to females.

However, these results had already been confirmed in previous research, so the real novelty of this research is that it provides information on whether the existing differences between males and females in anthropometric, psychological, and physical fitness variables increased or decreased when considering the level of physical activity, AMD, or weight status of the adolescents.

### 4.2. Differences between Male and Female Adolescents with Different Physical Activity Statuses in Anthropometric Variables, Psychological Variables, Physical Fitness Variables, AMD, and Weight Status: Can the Practice of Physical Activity Compensate for the Differences Found According to Gender in the Anthropometric and Derived Variables, Psychological State and Physical Fitness of the Adolescents?

The second specific objective of the study was to determine whether the practice of physical activity can compensate for the differences found according to gender in the anthropometric and derived variables, psychological state, and physical fitness of the adolescents. Regarding the comparison between AM vs. SM, and AF vs. SF, the results showed no significant differences between AM and SM, or between AF and SF, in the anthropometric variables analyzed. Previous research has shown contrary results, with differences found between active and inactive adolescents [74], which could be due to the fact that previous studies did not perform a separate analysis of AM and SM and AF and SF, so the introduction of males and females in the same comparison group, with the existing anthropometric differences between them [10], could be the reason for the discrepancy with respect to the present study. Regarding the absence of differences between active and sedentary adolescents in the anthropometric variables of the present investigation, a possible explanation could be that the COVID-19 pandemic led to an increase in weight and body fat of adolescents due to a positive energy balance, regardless of their previous level of physical activity [75]. These changes in anthropometric variables are far from being compensated by the practice of physical exercise, based on the results obtained in the present investigation, although these results should be contrasted in future research.

Regarding the psychological and physical fitness variables, it should be noted that significant differences were found between AF and SF in all the variables, while in the comparison between AM and SM, significant differences were only found in competence, autonomy, and VO_2_ max. These results partially follow the line of previous research [76] in that it was observed that the psychological benefits of physical activity practice differed according to gender. This finding is consistent with the results obtained, but it was also found that males received more psychological benefits from the practice of physical activity, which is contrary to the present results. This could be due to the fact that during the COVID-19 pandemic, although both males and females were affected in terms of life satisfaction and basic psychological needs, the effect was much greater in females [31] so that the practice of physical activity in the months following the pandemic could have had a more favorable impact on females than on males. As for the physical fitness tests, the hormonal and physical changes during adolescence could mitigate the differences between AM and SM in the variables related to power and speed. This may be because the changes that occur at the onset of puberty lead to an increase in muscle and bone mass in males, regardless of the practice of physical activity, while in females, the physical changes mainly occur in the distribution and accumulation of adipose tissue [2], with a smaller increase in production of muscle mass and strength. Therefore, as changes in muscle mass and strength are not as noticeable in female adolescents, the practice of physical activity could be more determinant for differences in their physical performance.

Another relevant result is that between males and females with the same level of physical activity practice, no differences were found in life satisfaction or the satisfaction of basic psychological needs between AM and AF, but the differences were significant between SM and SF, with SF scoring lower. These results partially follow the line of previous research, which indicated that males showed higher scores than females in psychological variables [11,27,77], as they agree in the case of sedentary adolescents analyzed in this research, but in the case of active adolescents, no differences were found. These results highlight the importance of physical activity for female adolescents to achieve an adequate psychological state. However, they also point to the importance of considering the physical activity practiced as a variable that could explain the lack of agreement between the differences found in previous research on the psychological state of male and female adolescents.

When comparing AM and SF, the differences between males and females were even more remarkable, following the results of previous studies in which males showed higher scores except for the variables related to fat mass and the sit-and-reach test [10]. However, what is novel in the present study is that when comparing AF and SM, no differences were found between the groups in fat mass or life satisfaction, which is especially relevant as it indicates the usefulness of physical activity in decreasing the differences between males and females in fat mass accumulation, as well as in increasing the life satisfaction of adolescent females. These results corroborate the second hypothesis (H2), in that the regular practice of physical activity of females will compensate for the differences in anthropometric and derived variables, psychological state, and physical fitness with respect to males.

### 4.3. Differences between Male and Female Adolescents with Different AMD in Anthropometric Variables, Psychological Variables, Physical Fitness Variables, Physical Activity Level, and Weight Status: Can an Optimal AMD Compensate for the Gender Differences Found in the Anthropometric and Derived Variables, Psychological State and Physical Fitness of the Adolescents?

The third specific objective of the present study was to analyze whether optimal AMD can compensate for the gender differences found in the anthropometric and derived variables, psychological state, and physical fitness of the adolescents. The results with respect to the anthropometric variables and the physical fitness tests showed that the differences were not significant in either the comparison of ODM and PDM or ODF and PDF. These results are similar to those from previous research in adolescents, in which no significant differences were observed between poor and optimal AMD in both males and females groups [78]. This could be due firstly to the fact that the adoption of a healthy diet alone does not generate as many changes in anthropometric and derived variables, and physical capacities, as following a diet program combined with regular physical activity in the adolescent population [79] and secondly to the fact that the combination of adolescents with poor AMD and need to improve AMD to obtain a sample size comparable to that of adolescents with optimal AMD, could influence the statistical analysis.

However, the results showed that both ODM and ODF had a better level of physical activity and psychological state than PDM and PDF, respectively, with the differences being significant in competence in the case of females and relatedness in the case of males. Regarding the level of physical activity, the results were similar to those from previous research, in which adolescents with a higher AMD showed a higher level of physical activity [80]. This could be because one of the main determinants for adopting other healthy lifestyle habits during adolescence is the practice of physical activity [81]. With respect to psychological state, no previous research has analyzed differences in the satisfaction of basic psychological needs according to AMD in the adolescent population. The results of the present investigation seem to indicate that ODM and ODF showed greater satisfaction of basic psychological needs, which could be because ODM and ODF also showed a higher level of physical activity practice than PDM and PDF, but future studies in this area are needed to draw definitive conclusions.

When comparing PDM and PDF, and ODM and ODF, the differences were significant in anthropometric and physical fitness variables, similar to previous research that studied the differences between males and females [10]. However, in the psychological variables, only significant differences were found between PDM and PDF in the competence variable. The results of the present study show that although the psychological state of the adolescents differed according to the AMD in both the male and female groups, the differences in the psychological state disappeared when comparing males and females with optimal AMD. Thus, there does not seem to be differences between genders if the level of AMD is optimal. No previous research is known to have analyzed the differences in the psychological state of adolescents according to AMD and considering gender differences, but a possible explanation could be that male and female adolescents with greater AMD have better physical and psychological well-being, better family relationships, and autonomy support [82], with these factors being related to the satisfaction of basic psychological needs and life satisfaction in adolescents [83,84].

It should be noted that when comparing PDM and ODF, PDF and ODM, the differences in the anthropometric and physical fitness variables continued to be greater in males, except for the variables related to body fat and the sit-and-reach test. A relevant result is that the differences were only significant in the psychological variables when comparing PDF and ODM, with the score of ODM being higher, but not when comparing PDM and ODF. These results could be explained by the fact that AMD is not the only influential factor in the psychological state of adolescents, with physical activity also playing a determining role [85]. As shown in previous research, physical activity is higher in males [7], which could be the reason why ODM showed statistically significant differences in psychological variables as compared to PDF but not ODF with respect to PDM, with future research necessary to corroborate the influence of these variables on the psychological state of adolescents. These results allow us to reject the third research hypothesis (H3), as an optimal AMD in females did not compensate for the existing differences with males in anthropometric and derived variables, psychological state, and physical fitness.

### 4.4. Differences between Male and Female Adolescents with Different Weight Statuses in Anthropometric Variables, Psychological Variables, Physical Fitness Variables, Physical Activity Level, and AMD: Can the Maintenance of an Adequate Weight Status Compensate for the Differences Found According to Gender in the Anthropometric and Derived Variables, Psychological State and Physical Fitness of the Adolescents?

The fourth specific objective was to establish whether the maintenance of an adequate weight status can compensate for the differences found according to gender in the anthropometric and derived variables, psychological state, and physical fitness of the adolescents. When comparing OWM and OWF with NWM and NWF, the results showed higher scores in the anthropometric variables related to fat mass and muscle mass in the OWM and OWF. As for the physical fitness variables, the OWM and OWF adolescents showed a lower score in cardiorespiratory capacity and CMJ, but it should be noted that in the OWM group, the score was higher as compared to the NWM group in handgrip, while in the 20-m sprint test, the NWM group scored better than the OWM one. Previous research has shown similar results [86]; however, the novelty of the present article lies in the fact that only in the male group were there differences in handgrip and 20-m sprint between normal weight and overweight/obese individuals. This could be because these tests assess strength and speed. These aspects that develop differently in adolescent males and females due to sexual dimorphism [2] or the differences in sports participation between adolescent males and females [42].

Regarding the psychological variables, the differences were not significant when comparing NWM and OWM or NWF and OWF. These results are contrary to those found in previous research, in which overweight adolescents showed less satisfaction with life and suffered more social isolation and bullying [87]. This could indicate that weight status alone is not such a determining factor in the psychological state of adolescents, with other aspects acquiring greater relevance. In this regard, the practice of physical activity was important, as shown in previous studies [88]. The decrease in physical activity levels caused by peer teasing for being overweight/obese [89] could be the real factor affecting the psychological state. However, this could mainly be due to the reduction in physical activity and not as much to being overweight/obese.

It is worth noting that when comparing NWM and NWF, and OWM and OWF, males continued to show higher scores in the level of physical activity performed, in the physical fitness, and anthropometric variables, except for fat mass, the sum of three skinfolds, and the sit-and-reach test, both in the normal weight and overweight/obese groups, as shown in previous research [7,10]. Regarding the differences between groups, hip girth was significantly higher in the NWF than in the NWM but not in the overweight/obese group. These results are similar to those found in previous research, in which females had a greater accumulation of peripheral fat in the limbs or hips [90,91]. However, in obese individuals, the hypothalamic-pituitary-adrenal axis may be overly sensitive, breaking the balance between the lipogenic and lipolytic effects of the hormones related to fat accumulation, cortisol, and insulin may be the reason why no differences were found between OWM and OWF [5,90].

Regarding the differences in psychological variables in the normal weight and overweight/obese groups, the OWF showed significantly lower scores in life satisfaction, while the NWF showed a lower score in competence compared to OWM and NWM, respectively. These results are similar to those found in previous studies, in which females showed worse life satisfaction and competence than males [11,27]. A possible explanation could be that adolescence is a crucial stage in which adolescents are prone to teasing and taunting by their peers, with this situation more evident in females, who are more likely to suffer verbal-relational victimization, which hinders their participation in certain social activities and complicates their integration with the rest of their peers [89,92], thereby affecting their psychological state. The relevance of these results lies in the fact that adolescent females showed a lower satisfaction with life and basic psychological needs than males, providing more scientific evidence in an area that was not made clear in previous research.

The NWF scored significantly lower than NWM in competence. These results are contrary to those found in previous studies, in which females showed higher values in satisfying all basic psychological needs [93]. It must be underlined that factors such as the practice of physical activity are determinants for the satisfaction of basic psychological needs, and thus competence could be affected by this [94]. The results of the present research show that adolescent males have a significantly higher level of physical activity than females, being considered active, while females are considered inactive [52], which could be the reason why NWF showed lower competence scores than NWM. Future research in this area is necessary to establish the importance of physical activity in the satisfaction of competence in adolescent males and females.

With respect to the comparison between NWF and OWM, and NWM and OWF, it should be noted that there were significant differences in the anthropometric and physical fitness variables, although no significant differences were found in any of the comparisons in the psychological variables analyzed. Regarding the anthropometric variables, the comparison between OWM and NWF showed significant differences in all the variables analyzed, with the OWM showing the highest scores. However, in the comparison of OWF and NWM, OWF showed higher scores in all variables related to fat mass, but NWM showed higher muscle mass. Furthermore, no significant differences were found in corrected girths between OWF and NWM, which could be because overweight/obese adolescents develop greater muscle and bone mass [95], which could compensate for the differences normally found in muscle mass between males and females caused by sexual dimorphism. The absence of differences with respect to psychological variables provides further evidence of the lack of relevance of weight status in the psychological state of the adolescent population. With respect to the physical fitness variables, it is noteworthy that the CMJ test showed no significant differences between NWF and OWM, which could be due to the lower performance of OWM in this test, as they had to move more body mass. These results allow us to refute the fourth research hypothesis (H4), which stated that the maintenance of normal weight in females would compensate for the differences in anthropometric and derived variables, psychological state, and physical fitness with respect to males.

Based on the results obtained, which respond to the four specific objectives and the general objective, i.e., to analyze whether the practice of physical activity, optimal AMD, and adequate weight status can compensate for the differences found between adolescent males and females in anthropometric and derived variables, psychological state, and physical fitness, the practical applications of the present study highlight that the promotion of physical activity is fundamental for the entire adolescent population. However, it is even more relevant for females than for males for the improvement of physical fitness and psychological state, as it is a key aspect for the reduction of the differences found between males and females and may even compensate for the differences found based on gender in the accumulation of fat mass and life satisfaction. Regarding AMD, it is noteworthy that there were differences between males and females with low AMD but not between males and females with optimal AMD, with this being a relevant factor to consider for the promotion of healthy habits in this population, as the psychological differences between genders disappear if optimal AMD is achieved. With respect to weight status, it is noteworthy that females, both normal weight and overweight/obese, showed lower scores in various psychological variables than males, although no differences were found when comparing NWF and OWF, so these differences may not be due to weight status alone, but to other variables such as physical activity as well. Therefore, although the promotion of the Mediterranean diet and the maintenance of healthy weight status is important for the adolescent population, physical activity seems to play a more decisive role, so programs aimed at the acquisition of healthy habits, such as AMD, should consider the inclusion of sports practice, as it seems to facilitate the acquisition of the rest of the healthy habits, and to increase the benefits obtained through these programs.

This study is not free of limitations. Despite selecting four different geographical areas, the sample was selected conveniently in the educational centers where access was allowed. By dividing the sample into subgroups such as degree of AMD or weight status, the sample sizes were reduced, and although they were similar between males and females, they represented a small part of the total sample. With respect to variables such as the level of physical activity or AMD, these were calculated based on the adolescents’ scores on the questionnaires that collected information on their nutritional and sports habits in recent days, and this aspect must be considered. The maturational status of adolescents should be considered in future research, as it could influence the results obtained, as certain behavioral changes in healthy habits occur as a function of maturation. In addition, weight status was established according to the BMI classification established by the World Health Organization, which must be considered, as it could make it difficult to compare with other research studies using another classification. On the other hand, the use of BMI does not allow differentiation between the components of body composition (muscle mass or fat mass). This should be taken into account, as some adolescents with a high level of muscle mass could be included in the overweight/obese group, although this is not very common.

## 5. Conclusions

To conclude, the differences between AM-SM and AF-SF were not significant in the anthropometric variables, while in the physical fitness and psychological variables, the differences were significant between AF and SF in all variables, while for the comparison between AM and SM, significant differences were only found in competence, autonomy, and VO_2_ max. When comparing males and females with the same level of physical activity (AM-AF, SM-SF), it is important to note that in the psychological variables, differences were only found between SM and SF, with SF scoring lower. Regarding the comparison between males and females with different levels of physical activity (AM-SF, AF-SM), the differences were even more evident in all the variables analyzed between AM and SF, although no differences were found between AF and SM in fat mass or life satisfaction.

Regarding AMD, no differences were found between ODM-PDM and ODF-PDF in the anthropometric and physical fitness variables, although differences were found in the psychological variables, with ODM and ODF scoring higher. In the comparison between PDM-PDF and ODM-ODF, the differences were significant in the anthropometric and physical fitness variables in both groups but only in the psychological variables between ODM-ODF. Another novel aspect is that in the psychological variables, the comparison between PDF-ODM showed significant differences, with higher values in ODM, although no differences were found between PDM-ODF.

According to weight status, the OWM and OWF showed higher scores in the anthropometric variables, and lower performance in cardiorespiratory capacity and CMJ. However, only the OWM showed higher performance in handgrip, while the NWM did so in the 20-m sprint. There were no significant differences in psychological variables between NWM-OWM nor between NWF-OWF. When comparing males and females with the same weight status (NWM-NWF and OWM-OWF), differences were more evident in anthropometric and physical fitness variables between genders, while in psychological variables, OWF and NWF showed lower values than males. As for the comparison between NWF-OWM and NWM-OWF, the psychological variables showed no significant differences, while in physical fitness, it is noteworthy that there were no differences between NWF and OWM in the CMJ. Regarding the anthropometric variables, the OWM showed higher scores than the NWF in all the variables analyzed, with this being similar in the OWF with respect to the NWM, except in muscle mass, in which the NWM obtained higher values, and in corrected girths, in which there were no differences between OWF and NWM.

## Figures and Tables

**Figure 1 children-09-01926-f001:**
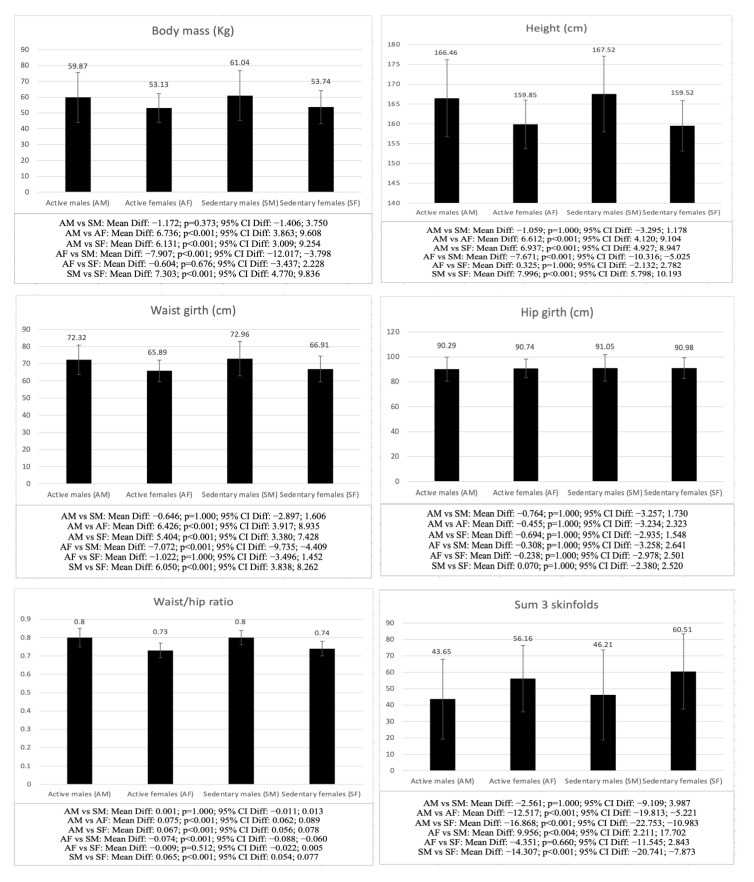
Differences in anthropometric, psychological, and physical fitness variables between males and females with different physical activity statuses.

**Figure 2 children-09-01926-f002:**
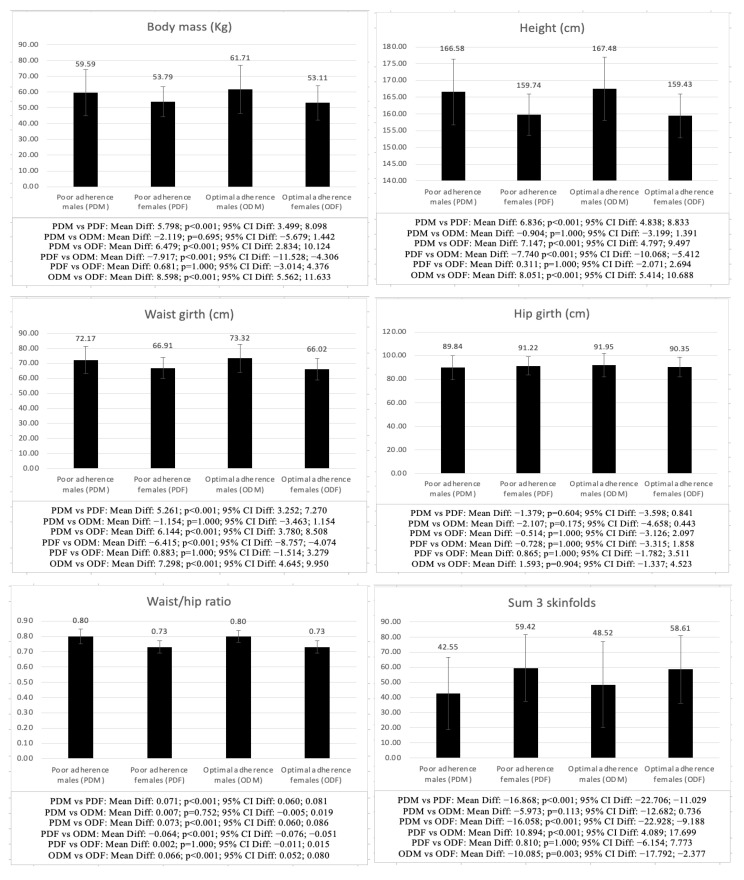
Differences in psychological, anthropometric, and physical fitness variables between males and females with different adherence to the Mediterranean diet.

**Figure 3 children-09-01926-f003:**
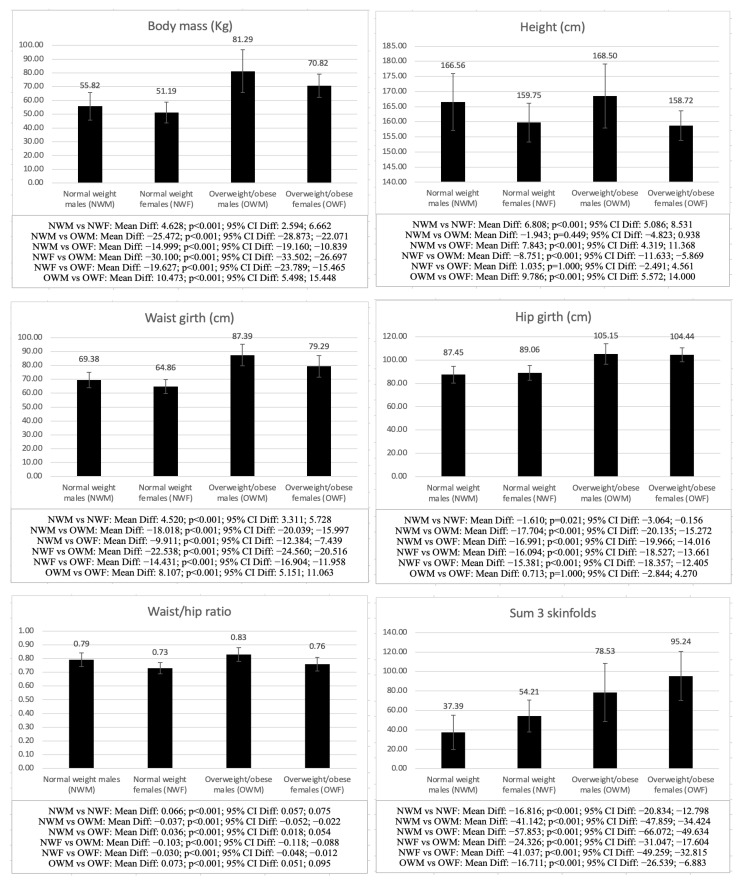
Differences in psychological, anthropometric, and physical fitness variables between males and females with different weight statuses.

**Table 1 children-09-01926-t001:** Dependent and independent variables of the study.

Variable Type	Construct	Variable
Independent	Physical activity	Physical activity status (active or sedentary)
	Diet	Adherence to Mediterranean Diet (AMD, poor adherence, or optimal adherence)
	Weight status	Weight status (normal weight or overweight/obese)
Dependent	Anthropometric and derived variables	Body mass (kg)
		Height (cm)
		BMI (kg/m^2^)
		Waist girth (cm)
		Hip girth (cm)
		Waist/hip ratio
		Sum 3 skinfolds (cm)
		Corrected arm girth (cm)
		Corrected thigh girth (cm)
		Corrected calf girth (cm)
		Fat mass (%)
		Muscle mass (kg)
	Psychological variables	Life satisfaction (punctuation)
		Competence (punctuation)
		Autonomy (punctuation)
		Relatedness (punctuation)
	Physical fitness variables	VO_2_ max. (mL/kg/min)
		Handgrip right arm (kg)
		Handgrip left arm (kg)
		Sit-and-reach (cm)
		CMJ (cm)
		20-m-sprint (s)
	Physical activity	Physical activity level (punctuation)
	Diet	Adherence to Mediterranean diet (punctuation)

BMI: body mass index; AMD; adherence to Mediterranean diet; CMJ: countermovement jump; VO_2_ max.: maximum oxygen consumption.

**Table 2 children-09-01926-t002:** Differences in psychological, anthropometric, and physical fitness variables between males and females and the influence of physical activity status, AMD, and weight status as covariables in the differences depend on gender.

Variable	Descriptors (M + SD)					GenderxPhysical Activity Status			GenderxBMI			GenderxAMD		
	Males (n = 404)	Females (n = 387)	F	*p*	ES (η^2^)	F	*p*	ES (η^2^)	F	*p*	ES (η^2^)	F	*p*	ES (η^2^)
Body mass (Kg)	60.36 ± 14.85	53.54 ± 10.00	53.286	<0.001	0.067	27.083	<0.001	0.068	313.114	<0.001	0.457	27.667	<0.001	0.069
Height (cm)	166.91 ± 9.66	159.63 ± 6.28	146.409	<0.001	0.164	73.391	<0.001	0.165	73.579	<0.001	0.165	73.281	<0.001	0.165
Waist girth (cm)	72.59 ± 9.18	66.58 ± 7.15	98.236	<0.001	0.117	50.005	<0.001	0.119	489.182	<0.001	0.568	49.351	<0.001	0.117
Hip girth (cm)	90.61 ± 10.10	90.90 ± 7.99	0.194	0.659	0.001	0.382	0.683	0.001	277.267	<0.001	0.427	1.227	0.294	0.003
Waist/Hip ratio	0.80 ± 0.05	0.73; 0.04	454.176	<0.001	0.379	227.505	<0.001	0.380	276.630	<0.001	0.427	228.429	<0.001	0.381
Sum 3 skinfolds (cm)	44.72 ± 25.71	59.12 ± 22.16	66.624	<0.001	0.082	35.118	<0.001	0.086	271.552	<0.001	0.422	35.238	<0.001	0.087
Corrected arm girth (cm)	22.58 ± 3.07	20.00 ± 2.28	163.373	<0.001	0.180	81.945	<0.001	0.181	191.159	<0.001	0.340	81.605	<0.001	0.180
Corrected thigh girth (cm)	41.96 ± 5.17	38.27 ± 3.84	118.886	<0.001	0.138	61.506	<0.001	0.142	138.021	<0.001	0.271	60.491	<0.001	0.140
Corrected calf girth (cm)	30.41 ± 2.97	28.21 ± 3.17	94.370	<0.001	0.113	47.905	<0.001	0.114	75.962	<0.001	0.170	47.183	<0.001	0.113
Fat mass (%)	20.31 ± 11.30	25.49 ± 8.28	50.300	<0.001	0.063	27.210	<0.001	0.068	262.921	<0.001	0.414	27.395	<0.001	0.069
Muscle mass (kg)	22.09 ± 4.79	15.62 ± 2.83	490.082	<0.001	0.397	245.723	<0.001	0.398	363.749	<0.001	0.495	245.806	<0.001	0.398
Life satisfaction	18.37 ± 4.26	17.50 ± 4.78	7.016	0.008	0.009	8.739	<0.001	0.023	3.523	0.030	0.009	8.473	<0.001	0.022
Competence	27.97 ± 6.73	25.81 ± 7.07	18.211	<0.001	0.024	45.339	<0.001	0.109	9.432	<0.001	0.025	18.706	<0.001	0.048
Autonomy	26.22 ± 6.24	25.15 ± 6.29	5.424	0.020	0.007	15.606	<0.001	0.040	2.732	0.066	0.007	9.624	<0.001	0.025
Relatedness	25.04 ± 6.00	24.46 ± 6.03	1.759	0.185	0.002	10.353	<0.001	0.027	1.030	0.357	0.003	9.485	<0.001	0.025
VO_2_ max.	42.27 ± 5.60	36.89 ± 4.41	209.870	<0.001	0.220	138.551	<0.001	0.272	133.079	<0.001	0.264	107.825	<0.001	0.225
Handgrip right arm	30.52 ± 9.02	22.71 ± 4.89	211.228	<0.001	0.221	106.201	<0.001	0.222	123.703	<0.001	0.250	105.690	<0.001	0.221
Handgrip left arm	28.48 ± 8.17	21.00 ± 4.22	241.497	<0.001	0.245	120.912	<0.001	0.246	137.521	<0.001	0.270	121.681	<0.001	0.247
Sit-and-reach	12.81 ± 7.51	19.20 ± 8.81	114.365	<0.001	0.133	60.241	<0.001	0.140	57.193	<0.001	0.133	57.135	<0.001	0.133
CMJ	26.38 ± 7.46	20.77 ± 5.01	142.791	<0.001	0.161	78.631	<0.001	0.175	94.606	<0.001	0.203	71.370	<0.001	0.161
20-m sprint	3.72 ± 0.54	4.15 ± 0.43	144.494	<0.001	0.163	78.866	<0.001	0.175	81.831	<0.001	0.181	72.158	<0.001	0.163
Physical activity level	2.83 ± 0.63	2.48 ± 0.61	59.888	<0.001	0.075	-	-	-	30.910	<0.001	0.078	50.447	<0.001	0.121
AMD	2.23 ± 0.66	2.26 ± 0.63	0.377	0.540	0.001	22.097	<0.001	0.053	0.371	0.690	0.001	-	-	-
BMI (kg/m^2^)	21.53 ± 4.15	21.03 ± 3.61	3.021	0.083	0.004	1.623	0.198	0.004	-	-	-	2.340	0.097	0.006

BMI: body mass index; AMD; adherence to Mediterranean diet; CMJ: countermovement jump; VO_2_ max.: maximum oxygen consumption.

## Data Availability

The data presented in this study are available on request from the corresponding author. The data are not publicly available because they contain information that could compromise the privacy of research participants but are available from the corresponding author upon reasonable request.

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
