# Peer review of "The Importance of Healthy Habits to Compensate for Differences between Adolescent Males and Females in Anthropometric, Psychological and Physical Fitness Variables"

_children, 2022, doi:10.3390/children9121926_

Round 1

Reviewer 1 Report

This work describes differences in a group of adolescents regarding to physical activity, Mediterranean diet adherence, VO2 and other variables. Among several variables analyzed, it is well characterized that physical activity is one of the most important thing that thins differences between males and females. 

This work is well described, just few grammatical errors.

Author Response

- This work describes differences in a group of adolescents regarding to physical activity, Mediterranean diet adherence, VO2 and other variables. Among several variables analyzed, it is well characterized that physical activity is one of the most important thing that thins differences between males and females. This work is well described, just few grammatical errors.

+ Dear reviewer, thank you very much for agreeing to review this article and for your assessment. We have sent the article to an expert translator for review and correction of possible grammatical errors. Thank you again.

Reviewer 2 Report

An interesting work. The manuscript presents interesting research and is appropriate for the journal.
The abstract is clear and to the point, stressing both the specific application and the generic aspects of the work.
The description of the experimental methodology, the presentation of results, and the discussion of these are clear and rigorous.
From a scientific point of view, the paper is interesting; is suitably prepared for publication in the Children Journal.
The general idea of the paper is appealing. The formal level of the manuscript is followed according to the journal template. The content of the manuscript has an adequate and very good explanatory value.

Author Response

- An interesting work. The manuscript presents interesting research and is appropriate for the journal.

+ Thank you very much for your review.

- The abstract is clear and to the point, stressing both the specific application and the generic aspects of the work.

+ Thank you very much for your comment.

- The description of the experimental methodology, the presentation of results, and the discussion of these are clear and rigorous.

+ Thank you very much for your contribution.

- From a scientific point of view, the paper is interesting; is suitably prepared for publication in the Children Journal.

+ Thank you very much.

- The general idea of the paper is appealing. The formal level of the manuscript is followed according to the journal template. The content of the manuscript has an adequate and very good explanatory value.

+ Dear reviewer, thank you very much for taking the time to review this manuscript and agreeing to review it. Thank you for your assessment of this article.

Reviewer 3 Report

GENERAL COMMENTS

The aim of this paper was to examine the practice of physical activity, the Mediterranean diet, and adequate weight status for the differences found between adolescent males and females in anthropometric variables, psychological state, and physical fitness. Although this article addresses an interesting topic, many issues should be addressed before publication. Firstly, the manuscript requires proofreading by a native English proofreader. The use of sentence structures and grammar was hard to understand.

SPECIFIC COMMENTS

ABSTRACT

The abstract is missing of method and results.  

INTRODUCTION
The introduction needs major revision and clarification. First, the aim of the study is not clear. The research manuscript should answer a specific research question, which is lacking in this study. Although there are hypotheses stated, it was not clear whether it relates to boys or girls? Also, the way the authors build up their introduction does not lead to the research question. Although much of the necessary information regarding the background is already briefly written down, mixing between adults, adolescents and children. The authors should restructure their introduction, explaining why their research is important. Why should the difference between boys and girls on body fat, flexibility, psychological state and physical fitness? More importantly, this should lead to a clear research question rather than hypothesis.

METHODS
The methods section needs major revision. I am very confused. I think the sample itself, which is non-probability, is problematic, as it might create a ceiling effect as the participants are from the same group. How can it be related to another country, districts or even provinces? Maybe can lower the scope of the study?

2. Please provide an ethical approval code as dealing with vulnerable children.

3. I think this study's sample size does not have sufficient power. Normally, we use 95% confidence interval; please justify why using 99%? What is the effect size of the study? Besides, why was the calculated sample is 750 adolescents but collected 791 adolescents? Is it because of the dropout rate? Besides, are the boys and girls similar in age?

4. Why choose the psychological variables for life satisfaction, competence, autonomy and relatedness? Should the authors explain more clearly the theory of planned behaviour? Or motivational aspect?

5. Is the questionnaires used in this study validated for the Spanish version and for adolescents? It was stated for the SWLS questionnaire but not for the rest.

6. Why use speed measures rather than cardiorespiratory measures like VO2max such as the Bleep test?

RESULTS
The results section needs major revision. Including a research question would probably help the authors structure their results. The authors should amend Table 2 for easier comparison and explanation.

- Why suddenly did the boys sample become 359 from the 404 collected samples? However, female is the same? Please justify.

Figures 1, 2 and 3 are  too small and hard to give any meaning.

DISCUSSION
In the discussion section, the authors should further discuss their findings and the implication of these findings. They should also discuss their findings in more depth. The authors also discuss many topics that are not related to the results. It is too long and not focused on the results. In addition, they describe many studies in great detail which is not necessary for the discussion. The limitation is also not well thought out. Please revise.

Thank you.

Author Response

- The aim of this paper was to examine the practice of physical activity, the Mediterranean diet, and adequate weight status for the differences found between adolescent males and females in anthropometric variables, psychological state, and physical fitness. Although this article addresses an interesting topic, many issues should be addressed before publication.

+ Dear reviewer, thank you very much for agreeing to review the article and allowing us to improve it with your contributions.

- Firstly, the manuscript requires proofreading by a native English proofreader. The use of sentence structures and grammar was hard to understand.

+ Thank you very much. The article has been sent to an expert translator to facilitate understanding and avoid grammatical errors.

SPECIFIC COMMENTS

- ABSTRACT: The abstract is missing of method and results.

+ Thank you very much for your comment. Information regarding the method and results has been included, providing more information on the measurement environment and the statistical values obtained.

INTRODUCTION
- The introduction needs major revision and clarification. First, the aim of the study is not clear.

+ Thank you for your input. A first general objective has been included, which gives rise to the research, for which four specific objectives are postulated, in the hope that this will make it clearer.

- The research manuscript should answer a specific research question, which is lacking in this study.

+ Thank you very much for your contribution. The research question that gives rise to the research objectives has been included.

- Although there are hypotheses stated, it was not clear whether it relates to boys or girls?

+  Thank you for your input. The assumptions have been modified to make it clear whether the change in physical activity, AMD or weight status would impact males or females.

- Also, the way the authors build up their introduction does not lead to the research question.

+ Dear reviewer, thank you very much for your great input. The order of the introduction has been modified, making clear first the importance of the independent variables alone for males and females, and subsequently the existing differences between males and females in anthropometric variables, psychological state, and physical condition. This allowed us to define the research question in a coherent way.

- Although much of the necessary information regarding the background is already briefly written down, mixing between adults, adolescents, and children. 

+ Thank you very much for your contribution. It has been specified throughout the introduction that previous studies have been conducted on adolescents, and in those that also included children, the information was presented separately, allowing us to select only the adolescents.

- The authors should restructure their introduction, explaining why their research is important.

+ Thank you very much for your contribution. The introduction has been restructured, giving rise to the reason for this research: to know if healthy habits are determinant enough to compensate for the differences between males and females in anthropometric and derived variables, psychological state and/or physical fitness.

- Why should the difference between boys and girls on body fat, flexibility, psychological state and physical fitness? More importantly, this should lead to a clear research question rather than hypothesis.

+ Thank you very much for your input. A paragraph has been included in the introduction explaining the reasons for these possible differences.

METHODS
1. The methods section needs major revision. I am very confused. I think the sample itself, which is non-probability, is problematic, as it might create a ceiling effect as the participants are from the same group. How can it be related to another country, districts or even provinces? Maybe can lower the scope of the study?

+ Dear reviewer, thank you for your input. Although the sampling is non-probabilistic, it has been specified that the sample belonged to four education centers located in completely different geographical areas of the Region of Murcia, resulting in a representative sample that would allow, at least, a comparison with other provinces in Spain.

  1. Please provide an ethical approval code as dealing with vulnerable children.

+ Thank you very much. The code of the ethics committee that approved the research has been specified.

  1. I think this study's sample size does not have sufficient power. Normally, we use 95% confidence interval; please justify why using 99%? What is the effect size of the study? Besides, why was the calculated sample is 750 adolescents but collected 791 adolescents? Is it because of the dropout rate? Besides, are the boys and girls similar in age?

+ Dear reviewer, thank you very much for your input. The average age of males and females has been included separately. Regarding the sample size, it has been specified that, following the methodology of previous research, the sample size was calculated based on the standard deviations of previous research with adolescents. The fact of using a 99% confidence interval, instead of 95%, is simply to make the sample size calculation more rigorous. Thus, with an error of 0.05, a sample of at least 750 students was required to be representative. Finally, it was possible to include 791 adolescents in the research, making the sample representative. 

  1. Why choose the psychological variables for life satisfaction, competence, autonomy and relatedness? Should the authors explain more clearly the theory of planned behaviour? Or motivational aspect?

+ Thank you very much for your contribution. Both psychological constructs were selected because of their relationship with healthy habits (physical activity, nutritional pattern, and weight status), and because of the scarce information in previous scientific literature on the role they may play in this relationship.

  1. Is the questionnaires used in this study validated for the Spanish version and for adolescents? It was stated for the SWLS questionnaire but not for the rest.

+ Thank you very much for your comment. The citations of the articles that validated each instrument in Spanish have been included.

  1. Why use speed measures rather than cardiorespiratory measures like VO2max such as the Bleep test?

+  Thank you very much for your comment. In the present investigation, different tests were used to measure cardiorespiratory capacity, strength, power, hamstring and lower back flexibility and speed, with the aim of globally assessing physical fitness. The test used for each assessment was specified.

RESULTS
- The results section needs major revision. Including a research question would probably help the authors structure their results. The authors should amend Table 2 for easier comparison and explanation.

+ Thank you for your comments. We have introduced the research question in each of the sections of the results and modified table 2, as well as its title, for better understanding.

- Why suddenly did the boys sample become 359 from the 404 collected samples? However, female is the same? Please justify.

+ Dear Reviewer. Thank you very much for your suggestion. We did not realize, when performing the analysis, that there were subjects lost by the system in some of the variables; for this reason, there were fewer subjects in the males and in the females, since not all were being counted in all the variables. This problem has been solved and the corresponding variable has been modified in this table.

- Figures 1, 2 and 3 are  too small and hard to give any meaning.

+ Dear Reviewer. Thank you for your input. We have included the figures in that size so as not to occupy an excessive number of sheets. However, the size is modifiable or, if you consider it convenient, we could leave only the graphs of the variables that showed statistically significant differences, thus reducing the number of graphs and allowing us to enlarge the size of the rest.

DISCUSSION
- In the discussion section, the authors should further discuss their findings and the implication of these findings. They should also discuss their findings in more depth. The authors also discuss many topics that are not related to the results. It is too long and not focused on the results. In addition, they describe many studies in great detail which is not necessary for the discussion.

+ Dear reviewer, thank you for your comment. We have shortened the discussion, leaving only what refers to the results obtained, and eliminating those contributions that deviated from them. In addition, we have introduced the headings of the "results" section to make it easier to follow the article.

- The limitation is also not well thought out. Please revise.
+ Thank you very much for your input. The limitations section has been expanded to include other limitations to be considered.

- Thank you.

+ Dear reviewer, thank you very much for agreeing to review the manuscript and for all the contributions made. We have tried to comply with all of them to substantially improve the quality of the manuscript.

Round 2

Reviewer 3 Report

The authors had successfully addressed all the comments as suggested with satisfaction. Thus, I recommend accepting this manuscript. Thank you.